# IPA-3: An Inhibitor of Diadenylate Cyclase of *Streptococcus suis* with Potent Antimicrobial Activity

**DOI:** 10.3390/antibiotics11030418

**Published:** 2022-03-21

**Authors:** Haotian Li, Tingting Li, Wenjin Zou, Minghui Ni, Qiao Hu, Xiuxiu Qiu, Zhiming Yao, Jingyan Fan, Lu Li, Qi Huang, Rui Zhou

**Affiliations:** 1State Key Laboratory of Agricultural Microbiology, College of Veterinary Medicine, Huazhong Agricultural University, Wuhan 430070, China; lht@webmail.hzau.edu.cn (H.L.); li.tingting@webmail.hzau.edu.cn (T.L.); zouwenjin@webmail.hzau.edu.cn (W.Z.); minghuini@webmail.hzau.edu.cn (M.N.); huqiao@webmail.hzau.edu.cn (Q.H.); xiuxiuq@webmail.hzau.edu.cn (X.Q.); yaozhiming@webmail.hzau.edu.cn (Z.Y.); fjy6168@webmail.hzau.edu.cn (J.F.); lilu@mail.hzau.edu.cn (L.L.); 2Cooperative Innovation Center of Sustainable Pig Production, Wuhan 430070, China; 3International Research Center for Animal Disease (Ministry of Science & Technology of China), Wuhan 430070, China; 4The HZAU-HVSEN Institute, Wuhan 430042, China

**Keywords:** IPA-3, cyclic diadenylate monophosphate, diadenylate cyclase, high-throughput screening, inhibitor, antimicrobial, *Streptococcus suis*

## Abstract

Antimicrobial resistance (AMR) poses a huge threat to public health. The development of novel antibiotics is an effective strategy to tackle AMR. Cyclic diadenylate monophosphate (c-di-AMP) has recently been identified as an essential signal molecule for some important bacterial pathogens involved in various bacterial physiological processes, leading to its synthase diadenylate cyclase becoming an attractive antimicrobial drug target. In this study, based on the enzymatic activity of diadenylate cyclase of *Streptococcus suis* (ssDacA), we established a high-throughput method of screening for ssDacA inhibitors. Primary screening with a compound library containing 1133 compounds identified IPA-3 (2,2′-dihydroxy-1,1′-dinapthyldisulfide) as an ssDacA inhibitor. High-performance liquid chromatography (HPLC) analysis further indicated that IPA-3 could inhibit the production of c-di-AMP by ssDacA *in vitro* in a dose-dependent manner. Notably, it was demonstrated that IPA-3 could significantly inhibit the growth of several Gram-positive bacteria which harbor an essential diadenylate cyclase but not *E. coli*, which is devoid of the enzyme, or *Streptococcus mutans*, in which the diadenylate cyclase is not essential. Additionally, the binding site in ssDacA for IPA-3 was predicted by molecular docking, and contains residues that are relatively conserved in diadenylate cyclase of Gram-positive bacteria. Collectively, our results illustrate the feasibility of ssDacA as an antimicrobial target and consider IPA-3 as a promising starting point for the development of a novel antibacterial.

## 1. Introduction

Antimicrobial resistance (AMR) has become a serious global issue threatening human and animal health [1,2,3,4]. At least 700,000 deaths are caused by antibiotic resistance worldwide every year [5], and it has been estimated that 10 million people may die from AMR annually by 2050 if no effective actions are taken [6]. One of the major reasons leading to AMR accumulation is the slowdown of novel antimicrobial drug development. In the past few decades, very few novel classes of antibiotics have been developed [7]. Therefore, the development of novel antimicrobial drugs is urgently needed.

Drug target identification is critical for novel antimicrobial drug development [8]. Traditional antibiotics generally work by interfering with the biosynthesis of the bacterial cell wall, DNA replication, protein synthesis, or the integrity of the cell membrane [9,10]. Recently, novel pathways have been proposed as promising targets for antimicrobial drug development. Bacterial proteases such as FstH and signal peptidases I and II are regarded as antimicrobial drug targets due to their critical roles in bacterial physiology [11]. Bacterial kinases, including histidine kinases and serine/threonine kinases, are considered attractive targets for novel antibacterial development, and inhibitors of these kinases have been screened which display antimicrobial activity [12]. In addition, other potential antimicrobial drug targets have been proposed, such as the β-barrel assembly machine (BAM) complex [13] and the bacterial SOS pathway [14]. Recently, cyclic dinucleotide (c-di-GMP, c-di-AMP, and cGAMP) signaling was revealed to have critical regulatory roles in bacterial physiology, and these molecules have been deemed as promising antimicrobial and anti-virulence drug targets [15,16].

c-di-AMP is an emerging second-messenger molecule predominant in Gram-positive *Firmicutes*, *Actinomycetes*, and *Mycobacteria* [17,18,19]. It is involved in various physiological processes, including but not limited to maintaining cellular potassium hemostasis and osmotic pressure, regulating the synthesis of the cell wall, monitoring DNA damage, and controlling biofilm formation [20,21,22]. Cellular c-di-AMP levels are precisely regulated by diadenylate cyclase and phosphodiesterase [23]. The deletion mutant of diadenylate cyclase in many species (e.g., *Staphylococcus aureus* and *Streptococcus pneumoniae*) can not be constructed under common culture conditions, indicating that c-di-AMP is an essential molecule [24,25]. Therefore, targeting diadenylate cyclase could be a promising strategy to develop novel antimicrobials.

*Streptococcus suis* is an important zoonotic pathogen causing serious public health issues and economic losses [26,27]. It causes a wide range of diseases in pigs, including meningitis, arthritis, and sepsis [4]. *S. suis* can also cause life-threatening diseases such as streptococcal toxic shock-like syndrome (STSLS) and meningitis in humans [28,29]. Vaccines are deemed as a valid strategy to prevent infectious diseases. Unfortunately, the multiple serotypes and sequence types of *S. suis* commonly result in vaccination failure [30,31]. Currently, antibiotics are extensively utilized to treat diseases caused by *S. suis*. However, the misuse of antibiotics causes the accumulation of antimicrobial resistance in *S. suis* [32,33,34,35]. Thus, developing novel antibiotics is of great importance in controlling *S. suis* infection.

In this study, we established a high-throughput approach to screen the inhibitors of the second messenger c-di-AMP synthase of *S. suis* (ssDacA). Subsequently, a drug library including 1133 compounds was subjected to testing for their ssDacA inhibition. One compound, IPA-3 (2,2′-dihydroxy-1,1′-dinapthyldisulfide), was identified as an effective ssDacA inhibitor, demonstrating potent inhibition against *S. suis* and other Gram-positive bacteria.

## 2. Results

### 2.1. Purification of ssDacA

Diadenylate cyclase of *S. suis* is a triple membrane-spanning protein, which has a C-terminal catalytic domain (residues 99 to 283) (Figure 1A). The catalytic domain (ssDacA) was expressed in *E. coli* and purified by affinity chromatography. The purified ssDacA was analyzed by SDS-PAGE, which demonstrated that the protein was successfully obtained (Figure 1B).

### 2.2. Determination of the Optimal Enzymatic Reaction Conditions for ssDacA

We next sought to establish a high-throughput assay to screen ssDacA inhibitors. The parameters for the enzymatic reaction were optimized. As ssDacA catalyzes the condensation of 2 ATP molecules into cyclic di-AMP (Figure 2A), its activity can be indicated by the consumption of ATP. We used Kinase Glo^®^ reagent to measure the presence of ATP in the ssDacA catalytic reaction. Firstly, the optimal ATP concentration was determined. As shown in Figure 2B, ATP at 100 µM in the enzymatic reaction exhibited the largest signal–noise ratio, which was used as the optimal ATP concentration. Next, the optimal ssDacA concentration was determined in which 100 µM of ssDacA was the lowest concentration that consumed the most ATP (Figure 2C). Finally, the optimal incubation time was determined, indicating that 2 h was the shortest time to obtain the largest signal–noise ratio in the presence of 100 µM ATP and 100 µM ssDacA (Figure 2D). Together, 100 µM ATP, 100 µM ssDacA, and 2 h incubation at 37 °C were used as the optimal condition for ssDacA activity assay. By using this condition, we calculated the Z-factor (a parameter for quality control for high-throughput screening [36,37]) as 0.67 (Figure 2E), indicating that the established assay was suitable for high-throughput screening.

### 2.3. Screening for ssDacA Inhibitors

By using the assay established above, a drug library containing 1133 compounds was applied to screen for ssDacA inhibitors (Figure 3A and Appendix A). The results revealed that IPA-3 (2,2′-dihydroxy-1,1′-dinapthyldisulfide) exerted an inhibition of 82.33% against ssDacA at 100 μM. The structure of IPA-3 is shown in Figure 3B. The enzymatic assay indicated that the half-maximal inhibitory concentration (IC_50_) of IPA-3 was 38.22 μM against ssDacA (Figure 3C). To further confirm the inhibitory activity of IPA-3 against ssDacA, the catalytic product c-di-AMP in the reaction was detected in the presence and absence of IPA-3 *in vitro* by high-performance liquid chromatography (HPLC). IPA-3 demonstrated the ability to inhibit the production of c-di-AMP by ssDacA in a concentration-dependent manner (Figure 3D).

### 2.4. Antimicrobial Activity of IPA-3

As diadenylate cyclase is an essential protein in several bacteria and believed to be an antimicrobial drug target, we subsequently tested the antimicrobial activity of IPA-3. Three bacteria strains including *S. suis* SC19 [38], *B. subtilis* WB800N [39], and *S. aureus* ATCC29213 were subjected to growth tests in the presence of different concentrations of IPA-3 ranging from 0 to 25 μM. The bacterial growth assay demonstrated that IPA-3 at 25 μM almost completely abolished the growth of these strains. Additionally, IPA-3 at 5 μM or 10 μM demonstrated growth inhibition (Figure 4A–C). Notably, IPA-3 was also demonstrated to inhibit the growth of antimicrobial-resistant bacterial strains including *E. rhusiopathiae* 13013 [40], *S. suis SS2041* [41], and *S. aureus* 1213M4A (Figure 4D–F). However, IPA-3 at different concentrations demonstrated no or non-lethal inhibition against *S. mutans* ATCC25175 (Figure 4G) and *E. coli* ATCC25922 (Figure 4H).

### 2.5. Potential Binding Mode

A simulated structure of ssDacA was generated by using the I-TASSER server (Figure 5A). IPA-3 was docked into the simulated 3D structure of ssDacA. The lowest binding energy (−8.84 kcal/mol) conformation revealed that IPA-3 interacts with ssDacA via hydrogen bonds and hydrophobic forces. In order to evaluate the binding mode of the complex (ssDacA–IPA-3), a molecular dynamics simulation was performed using GROMACS software. The root-mean-square deviation (RMSD) was introduced to monitor the fluctuations of the simulation process. It was shown in Figure 5B that the complex ssDacA–IPA-3 finally reached a stable and equilibrious state. Based on the optimized conformation with minimal energy, the residues in ssDacA that interact with IPA-3 include L141, D181, A196, L198, T212, and R213. The multiple amino acid alignments based on different bacterial diadenylate cyclase amino acid sequences revealed that most of these residues are relatively conserved (Figure 5C,D).

## 3. Discussion

The development of novel antibiotics is an effective strategy to tackle antimicrobial resistance. Recently, c-di-AMP has been revealed as an essential signal molecule in several important bacterial pathogens, making c-di-AMP synthase an attractive antimicrobial target. In this study, we developed a novel high-throughput assay to screen for ssDacA inhibitors, and IPA-3 was identified as a potent inhibitor showing inhibition against ssDacA enzymes as well as *S. suis* and several other bacteria. Our results provide a good starting point for further antimicrobial drug development.

c-di-AMP was originally discovered as an important signal molecule involved in DNA repair [42]. It was later demonstrated that c-di-AMP plays major roles in the regulation of physiological homeostasis connected with bacteria fitness [43,44], including maintaining cell wall homeostasis, regulating cellar metabolism, monitoring DNA integrity, influencing sporulation, and biofilm formation [20,45,46,47]. For instance, the genetic competence of *S. pneumoniae* is modulated by c-di-AMP, which further influences its antibiotic tolerance and environmental response [48]. c-di-AMP signaling can also affect normal physiological functions and impair the virulence of *E. faecalis* [49]. Furthermore, in *L. monocytogenes*, diminished c-di-AMP levels lead to declined growth of bacteria in macrophages, indicating that c-di-AMP is critical for establishing infection [46]. More importantly, c-di-AMP was reported as an essential second-messenger molecule. It has been reported that diadenylate cyclase cannot be deleted in *S. suis*, *S. aureus*, or *S. pneumoniae* under normal culture conditions [24,50]. Collectively, the synthesis of c-di-AMP in bacteria could be a promising antimicrobial target.

The usual method of screening for diadenylate cyclase inhibitors is based on coralyne associated fluorescence assay. On binding with c-di-AMP, the fluorescence of coralyne can be significantly enhanced and this can be performed to monitor c-di-AMP biosynthesis [51,52,53,54]. In this study, we established a novel method of detecting the activity of ssDacA by measuring the consumption of ATP, which shortened the screening cycle. To maximize the difference of signal to background, the important reaction parameters were optimized for the enzymatic reaction system. We consider the concentration of 100 μM ssDacA and 100 μM ATP incubated at 37 °C for 2 h to be the optimal enzymatic reaction parameters. Z-factor is a statistical parameter for estimating the signal dynamic range and the data variation of the high-throughput screening assay [55]. The Z-factor value of 0.67 demonstrated that this method of enzyme activity measurement could be used in a high-throughput screening assay.

So far, several inhibitors of diadenylate cyclase have been identified. A series of the *B. subtilis* DisA inhibitors such as bromophenol thiohydantoin, tannic acid, theaflavin-3′-gallate, theaflavin-3, 3′-digallate, and suramin were found by employing coralyne fluorescence assay [52,53]. Cordycepin triphosphate was also revealed as an inhibitor of *Thermotoga maritima* DisA [54]. However, antimicrobial evaluations of the inhibitors *in vitro* and *in vivo* were not carried out.

IPA-3 is an allosteric inhibitor of p21-activated kinase-1 (PAK-1) that plays an essential role in eukaryotic cell migration, proliferation, and gene transcription, and acts as an anti-tumor target [56]. IPA-3 was also found to have bioactivity against many cancer cells, such as metastatic prostate cancer cells and a variety of human leukemic cell lines [57]. Here, we report for the first time that IPA-3 is an inhibitor of diadenylate cyclase. In addition, IPA-3 at 25 μM can almost completely inhibit the growth of Gram-positive bacteria, including AMR strains harboring an essential diadenylate cyclase, but not *E. coli* which is devoid of diadenylate cyclase. However, it was revealed that the level of inhibition of IPA-3 against *S. mutans* ATCC25175 was much lower than against other Gram-positive bacteria tested. This was consistent with a previous report in which it was demonstrated that diadenylate cyclase was not completely vital for the growth of *S. mutans* [58].

As the 3D structure of DacA of *S. suis* remains to be resolved, a simulated structure was generated using the I-TASSER server. Subsequently, IPA-3 was docked into the simulated structure. The lowest-energy conformation was considered the binding mode between IPA-3 and the simulated structure of ssDacA. To further optimize the binding mode of the lowest-energy conformation of ssDacA–IPA-3, a molecular dynamics simulation was performed. In the optimized complex model, IPA-3 interacts with ssDacA via hydrogen bonds and hydrophobic forces. It was also reported that the key motifs for the enzymatic activity of ssDacA were DGA and RHR, which are relatively conserved in diadenylate cyclase among Gram-positive bacteria [59]. Moreover, the residues in ssDacA that bind to IPA-3 include the residues in DGA and RHR motifs, which can explain why IPA-3 presents a wide-spectrum inhibition against *S. suis*, *S. aureus*, *B. subtilis*, and *E. rhusiopathiae.*

Although IPA-3 is regarded as a drug-like compound, a previous study revealed that IPA-3 is toxic for human peripheral blood mononuclear cells [60]. In addition, we found that IPA-3 was poorly soluble in water, which is an obstacle to its direct use as an antibiotic. Therefore, further optimizations based on IPA-3 will be needed. For example, structural biological studies should be performed to give a more accurate binding model between IPA-3 and ssDacA. We recommend testing the bioactivity of IPA-3-derived compounds to provide more information for structure–activity relationship analysis. If more promising compounds are discovered, druggability studies such as absorption, distribution, metabolism, and excretion (ADME) analysis could be carried out.

In conclusion, we developed a high-throughput assay to screen for ssDacA inhibitors which identified IPA-3 as a potent inhibitor with bioactivity inhibiting the growth of *S. suis*, *S. aureus*, *B. subtilis*, and *E. rhusiopathiae*. Our results indicate that IPA-3 could be a promising candidate for further antimicrobial development.

## 4. Materials and Methods

### 4.1. Bacterial Strains and Drug Library

The bacterial strains used in this study are listed in Appendix A. *S. suis*, *S. aureus*, and *Erysipelothrix rhusiopathiae* were cultured with tryptic soy agar (TSA) or tryptic soy broth (TSB) medium supplemented with 10% fetal bovine serum (FBS). *Escherichia coli* and *Bacillus subtilis* were grown in lysogeny broth (LB) medium. *Streptococcus mutans* was cultured with brain heart infusion (BHI) medium. The kinase inhibitor library (HY-LD-000001801), containing 1133 compounds, was purchased from MedChemExpress (MCE), and the detailed information is listed in Appendix A. The library was supplied in 96-well plates of 10 mM stocks in DMSO and stored at −80 °C.

### 4.2. Protein Expression and Purification

The coding sequence of the catalytic domain of diadenylate cyclase ssDacA (99 AA to 283 AA) of *S. suis* was amplified from the *S. suis* SC19 genome using the primer pair ssDacA_cyto_-F/R and is listed in Appendix A. The PCR product was then cloned into the pET28a vector to generate the recombinant expression plasmid pET28a-*dacA_cyto_*, which was transformed into *E. coli* BL21 (DE3) competent cells. The expression of the protein was induced by the addition of 1 mM isopropyl-β-D-thiogalactopyranoside (IPTG) at 28 °C for 10 h. The His-tagged ssDacA was purified by affinity chromatography with the Ni-NTA column (GE Healthcare, Uppsala, Sweden, Cat#: 10271899).

### 4.3. High-Throughput Screening for ssDacA Inhibitors

The screening of ssDacA inhibitors was performed as follows. A reaction mixture (10 μL) containing ssDacA (at optimal concentration) and ATP (at optimal concentration) in the reaction buffer (50 mM Tris-HCl, pH 7.5, 10 mM MgCl_2_, 150 mM NaCl) was supplemented with 0.5 μL of each library compound or DMSO in 384-well black plates. The plate was incubated at 37 °C for 2 h. Then, 10 μL of Kinase Glo^®^ reagent was added to each well. After 10 min, the relative light unit (RLU) values were measured by using a microplate spectrophotometer. Percent inhibition was calculated as ((RLU_X_ − RLU_P_)/(RLU_N_ − RLU_P_)) × 100%, where RLU_X_ is the RLU value for a test treated with compound X, and RLU_P_ and RLU_N_ are the RLU values for the reaction mixture without the treatment of compound and the reaction mixture lacking ssDacA, respectively.

The parameters for the ssDacA enzymatic reaction were optimized as follows. The optimal ATP concentration was determined in the presence of 50 µM ssDacA with ATP concentrations of 0, 20, 40, 60, 80, and 100 µM. Similarly, the optimal concentration of ssDacA was determined in the presence of the optimal concentration of ATP. The optimal reaction time was then determined when optimal concentrations of ssDacA and ATP were present.

The Z-factor is a statistical parameter for estimating the signal dynamic range and the data variation of the high-throughput screening assay. The Z-factor was calculated as 1 − [(3 × SD_N_ + 3 × SD_P_)/(AVG_N_ − AVG_P_)] where SD_N_ and SD_P_ are the standard deviation of the relative light unit (RLU) values of the 100 negative wells lacking ssDacA and the 100 positive wells containing ssDacA under the condition of optimal reaction parameters, respectively; in addition, AVG_N_ and AVG_P_ are the average RLU values of the negative wells and the positive wells, respectively.

### 4.4. Determination of Half-Maximal Inhibitory Concentration (IC_50_) of IPA-3 against ssDacA

The reaction was performed in a 50 μL mixture containing 100 μM ssDacA in the reaction buffer supplemented with 1 μL of DMSO or compound IPA-3 with final concentrations ranging from 10 μM to 200 μM in a 96-well black plate. The percent inhibition was calculated as described above. The data were transformed to log scale and non-linear regression was performed with GraphPad Prism software (version 7) using the variable-slope 4-parameter model for enzyme inhibition to determine IC_50_.

### 4.5. High-Performance Liquid Chromatography Analysis

High-performance liquid chromatography (HPLC) was used to determine the concentration of c-di-AMP as previously described with minor modifications [19]. Briefly, 500 μL of reaction mixture containing 50 μM ssDacA and 100 μM ATP in the reaction buffer supplemented with different concentrations of IPA-3 or DMSO was incubated at 37 °C for 2 h. The reaction was then terminated by incubation at 100 °C for 10 min. The mixture was centrifuged at 12,000 rpm for 10 min to remove the denatured protein, and the supernatant was filtered and degassed. Following this, 20 μL of the supernatant was analyzed by reversed-phase HPLC on an RPC-18 column (250 mm × 4.6 mm, GL Sciences, Tokyo, Japan) using the Agilent 1260 Infinity II HPLC system with 10 mM ammonium acetate, pH 5.5 (Buffer A), and 100% methanol (Buffer B) as solvent. The column temperature was set to 25 °C and the flow rate was 0.7 mL/min. Samples were eluted using a linear gradient from 0 to 50% solvent B over 30 min. c-di-AMP was detected by measuring absorbance at 254 nm. c-di-AMP standard (BioLog, Bremen, Germany, Cat NO. C 088–01) was run in parallel.

### 4.6. Bacterial Growth Inhibition Assay

Cells of *S. suis* SC19, *S. suis* SS2041, *S. aureus* ATCC29213, *S. aureus* 1213M4A, *S. mutants* ATCC25175, *E. rhusiopathiae* 13013, *B. subtilis* WB800N, and *E. coli* ATCC25922 were grown to the mid-log phase in TSB-FBS, LB, or BHI, respectively, according to Section 4.1. The cells were then subcultured 1:100 into the corresponding medium supplemented with different concentrations of IPA-3 (MedChemExpress) in a 100-well plate. The plate was incubated at 37 °C with shaking and the growth was monitored using an automatic growth curve analyzer (Oy Growth Curves Ab Ltd., Helsingfors, Finland).

### 4.7. In Silico Docking

The 3D structure of the diadenylate cyclase domain of ssDacA was predicted using the I-TASSER server (https://zhanglab.ccmb.med.umich.edu/I-TASSER/, accessed on 9 July 2020) [61,62,63]. The homologous model of the diadenylate cyclase domain of ssDacA with IPA-3 was generated using Autodock4 software. The residues of the diadenylate cyclase domain interacting with IPA-3 were displayed using PyMOL (version 2.0.6.0). The amino acid sequences of diadenylate cyclase were aligned using MEGA version 6 and were presented by the ESPript 3.0 server [64].

### 4.8. Molecular Dynamics Simulation

All-atom molecular dynamics simulation was performed to optimize the ssDacA–IPA-3 complex model using GROMACS software (2021 version). An Amber99SB-ILDN force field was used to describe both ssDacA and IPA-3. The topology file of IPA-3 was generated using Antechamber and ACPYPE tools. In the simulation system, the complex was set in a dodecahedral solvation box with the boundary kept at a minimum distance of 1.5 nm from the complex surface, and then the TIP3P water model was selected and 7 Na^+^ were added to the complex to counteract the charge of the system, based on the VERLET method. The simulation system was optimized for energy minimization under the Amber99SB force field, and then NVT and NPT runs were carried out for pre-equilibration. Subsequently, a total of 50 ns simulation under Amber99SB force field with a 2 fs step size was run in an NPT ensemble for this system, in which the temperature was set to 300 K and the pressure was set to 1.01325 bar. During the molecular dynamics simulation, the energy of the system, the RMSD of protein structure, and small-molecule structure fluctuation were monitored.

### 4.9. Statistical Analysis

The data were analyzed by a two-tailed Student’s *t*-test in GraphPad Prism 7 software, with a *p*-value < 0.05 considered to be statistically significant.

## Figures and Tables

**Figure 1 antibiotics-11-00418-f001:**
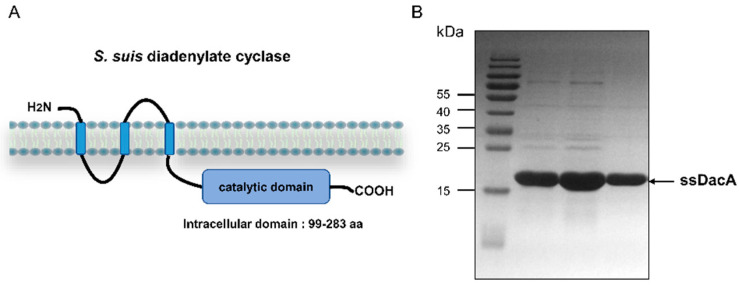
Purification of the catalytic domain of diadenylate cyclase of *S. suis*. (**A**) The predicted topology of *S. suis* diadenylate cyclase; (**B**) SDS-PAGE analysis of purified ssDacA.

**Figure 2 antibiotics-11-00418-f002:**
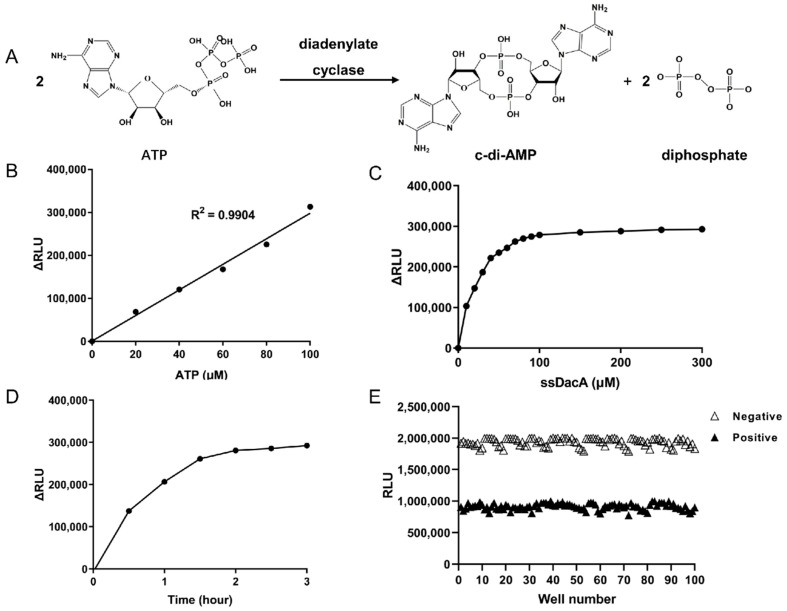
Optimization of parameters for the enzymatic reaction of ssDacA. (**A**) Biochemical reaction of diadenylate cyclase; (**B**) determination of the optimal ATP concentration. A reaction mixture (10 μL) containing 50 µM ssDacA with varied ATP concentrations of 0, 20, 40, 60, 80, 100 µM, in the reaction buffer (100 μM ssDacA, 50 mM Tris-HCl, pH 7.5, 10 mM MgCl_2_, 150 mM NaCl) in a 384-well black plate, was incubated at 37 °C for 2 h. Then, 10 μL of Kinase Glo^®^ reagent was added to each well. After 10 min, the relative light unit (RLU) values were measured using a microplate spectrophotometer. ΔRLU was calculated referring to wells containing the same amount of ATP but lacking ssDacA. (**C**) Determination of the optimal ssDacA concentration. The optimal concentration of ssDacA was determined in the presence of the optimal concentration of ATP as described above. (**D**) Determination of the optimal reaction time. The optimal reaction time was then determined when optimal concentrations of ssDacA and ATP were present as described above. (**E**) Determination of Z-factor. Enzymatic reactions with 100 replicates of positive wells containing ssDacA (▲) and the 100 replicates of negative wells lacking ssDacA (△) were carried out. The Z-factor was calculated as described in the Materials and Methods.

**Figure 3 antibiotics-11-00418-f003:**
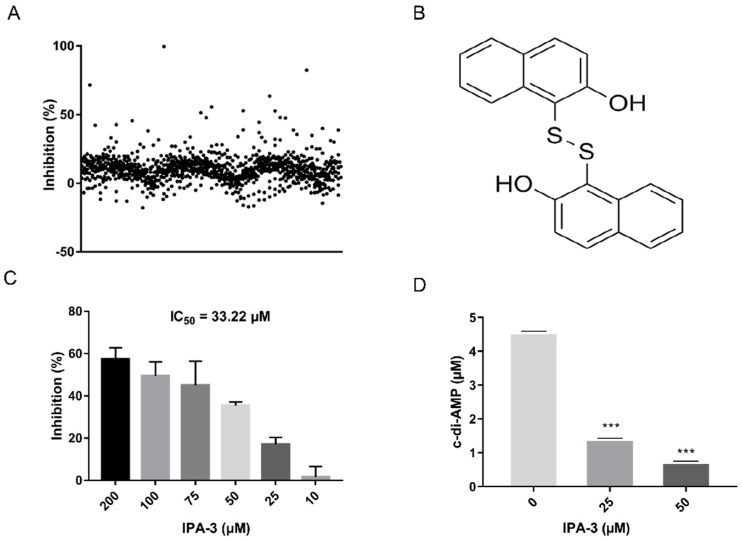
Screening for ssDacA inhibitors. (**A**) The scatter plot of the primary screening with the 1133 compounds; (**B**) the structure of IPA-3; (**C**) determination of IC_50_ of IPA-3 against ssDacA. Reactions containing 100 μM of ssDacA and varied concentrations of IPA-3 (10, 25, 50, 75, 100, 200 μM), were performed and the IC_50_ was calculated using the variable-slope 4-parameter model. The data presented are the means ± standard errors of the mean (*n* = 3). (**D**) Inhibition of IPA-3 on the production of c-di-AMP. Reactions containing 50 μM of ssDacA and varied concentrations of IPA-3 (0, 25, 50 μM) were performed in vitro. The produced c-di-AMP was quantified by HPLC. The data presented are the means ± standard errors of the means (*n* = 3). *** represents *p* value < 0.001.

**Figure 4 antibiotics-11-00418-f004:**
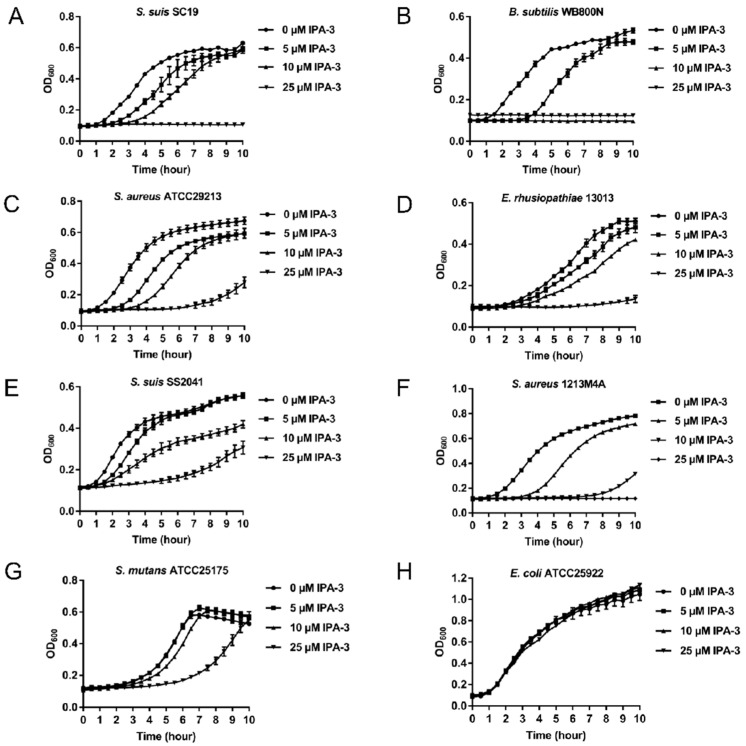
Antimicrobial efficacy of IPA-3. Cells of (**A**) *S. suis* SC19, (**B**) *B. subtilis* WB800N, (**C**) *S. aureus* ATCC29213, (**D**) *E. rhusiopathiae* 13013, (**E**) *S. suis* SS2041, (**F**) *S. aureus* 1213M4A, (**G**) *S. mutans* ATCC25175, and (**H**) *E. coli* ATCC25922 were subcultured from a culture grown overnight in an appropriate medium in the absence or presence of the indicated concentrations of IPA-3. The growth was monitored using an automatic growth curve analyzer. The data presented are the means ± standard errors of the means (*n* = 3).

**Figure 5 antibiotics-11-00418-f005:**
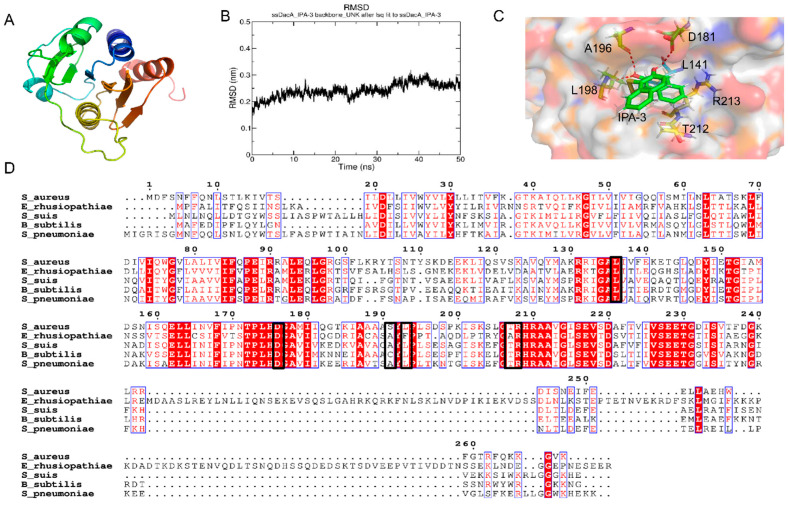
Analysis of the binding mode between IPA-3 and ssDacA. (**A**) The simulated 3D structure of ssDacA. The amino acid sequence of ssDacA was analyzed using the I-TASSER server. The image was generated by PyMOL software. (**B**) RMSD plot. The IPA-3 was docked to the simulated structure of ssDacA by using Autodock4 software and a molecular dynamics simulation was performed to further optimize the binding conformation using GROMACS software (2021 version), which generated the RMSD plot. (**C**) Optimized binding model between IPA-3 and ssDacA. The optimized conformation was taken from the stable and equilibrious time point in the molecular dynamics simulation. The protein–ligand 3D structure was generated using PyMOL software, where the green structure represents IPA-3 and the other structures represent the residues within the binding pocket of ssDacA. (**D**) Multiple sequence alignment of diadenylate cyclase from different bacteria. Multiple sequence alignment was generated by using MEGA version 6 software and the ESPript 3.0 server based on the amino acid sequence of diadenylate cyclase from each indicated bacterium. The amino acids involved in the interaction are shown in the black boxes.

## Data Availability

The data that support the findings of this study are available in the main manuscript and the Appendix A of this article.

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
