# Peer review of "IPA-3: An Inhibitor of Diadenylate Cyclase of Streptococcus suis with Potent Antimicrobial Activity"

_antibiotics, 2022, doi:10.3390/antibiotics11030418_

Round 1
Reviewer 1 Report
The authors report a new activity for a known compound. Since the compound was already reported as toxic to human, it won’t serve as an antibiotic. However, it has a value in future drug design research. Here are my suggestions for changes in the manuscript: Line 40 ‘Over 35,000 deaths are caused by antibiotic resistance in the United States every year.’ There’s no obvious reason why the authors specifically use the data for the United States. Perhaps they should instead provide global numbers (for instance by WHO), or data for the country that sponsored the research, as this should be more of their interest. Line 56 ‘Other potential antimicrobial drug targets include the β-barrel assembly machine (BAM) complex [13], bacterial SOS pathway [14], etc.’ this sentence should be modified to improve the style. Lines 71-77 More details should be provided about specific types of infections associated with S. suis. Methods Authors should provide more details about the drug library used in their screening. Line 195 ‘c-di-AMP was firstly found in Thermotoga maritime which was responsible for DNA repair’ this sentence should be modified to improve the style
Reviewer 2 Report
The manuscript entitled “IPA-3, an inhibitor against diadenylate cyclase of Streptococcus suis with potent antimicrobial activity” is dedicated to searching the novel antibiotics is an effective strategy to tackle antimicrobial resistance. Generally, the Manuscript is original and has a significance for the scientific community. Obtained results are reliable and supported by the data collected. The manuscript is easy to read and the arguments are described in a logical and understandable way.
In order to improve the manuscript, the following suggestion should be taken into account by the authors:
- The list of abbreviation should be added at the end of Manuscript.
- What stands the abbreviation IPA-3 for?
- At Figure 3 authors represent the structure of IPA-3. Are the other tested compounds belonging to naphthyl disulfide derivatives? Or the refer to other classes of organic compounds? There are no information about the other 1133 investigated compounds.
After this minor revision, I recommend the present manuscript for publication.
Reviewer 3 Report
The authors have selected a vast number of compounds to inhibit the growth of several Gram-positive bacteria. In respect to the in vitro analysis, the methodology used has successfully described how the process can be suppressed by the methods applied. However, the in silico analysis can be improved considerably. The binding site prediction by molecular docking is not innovative and do not brought sufficient results to consider the binding location where they have point out it was. In my opinion a Quantum mechanics could provide better description of the inhibitors interaction of the molecular interaction in the binding site. Molecular docking provides a superficial description of the molecular interaction. Therefore, this point must be improved in order to reach content quality that the journal requires.
Author Response
The authors have selected a vast number of compounds to inhibit the growth of several Gram-positive bacteria. In respect to the in vitro analysis, the methodology used has successfully described how the process can be suppressed by the methods applied. However, the in silico analysis can be improved considerably. The binding site prediction by molecular docking is not innovative and do not brought sufficient results to consider the binding location where they have point out it was. In my opinion a Quantum mechanics could provide better description of the inhibitors interaction of the molecular interaction in the binding site. Molecular docking provides a superficial description of the molecular interaction. Therefore, this point must be improved in order to reach content quality that the journal requires.
Response: Thanks for the reviewer’s constructive suggestion. We agree with the reviewer’s opinion that regular molecular docking may not provide the best results regarding the binding site. Quantum mechanics-based molecular docking is indeed a powerful approach as it can provide better accuracy in the description of protein-ligand interactions. However, unfortunately, due to computational limitations and time limits, we are unable to perform a quantum mechanics-based docking in this study.
To improve our current work, we performed a molecular dynamics simulation to optimize the binding mode between ssDacA and IPA-3. The RMSD data showed that the ssDacA-IPA-3 complex reached a stable state during the process of simulation (Please see Fig. 5). Most of the residues in ssDacA interacting with IPA-3 in the equilibrious conformation were the same as those described in our previous results. This suggests the binding site is reliable. We have included the molecular dynamics simulation results in the revised manuscript (Please see lines 174-182).

Reviewer 4 Report
Li et al. have reported IPA-3, an inhibitor against diadenylate cyclase of Streptococcus suis with potent antimicrobial activity.
Overall, the paper is scientifically sound, clear, concise without any major questionmarks from my end regarding the scientific validity of the data.
Revisions:
1) Most of the figures look blurry with significant amount of data so that needs to be fixed. Please include high quality figures that are clear and easy to see the details of the data
2) the authors mention that " a drug library containing 1133 compounds .." What is this library?Where is it? Can it be included as a supplementary info? Where did this 1133 compounds come from?
3) Any analytical HPLC after protein purification? Are the authors sure that the protein was purified to acceptable standards?
4) Bacterial growth inhibition assay: The authors need to include (at least cite) the full anti bacterial assays protocol! Additionally even though the authors mention assays against B. subtilis , this bacterial strain is not included in the protocol/methodology section.
5) It would be good if the authors could include a paragraph at the end of the conclusions section that suggests what would the future work include. There are so many papers on antimicrobial agents but a lot of them seem to be a dead end. It would be good if the authors could highlight a bit further what would the future work include? Why are these data more important than any other submitted papers?
Author Response
Li et al. have reported IPA-3, an inhibitor against diadenylate cyclase of Streptococcus suis with potent antimicrobial activity.
Overall, the paper is scientifically sound, clear, concise without any major question marks from my end regarding the scientific validity of the data.
Response: We appreciate the reviewer’s positive comments.
Revisions:
Point #1: Most of the figures look blurry with significant amount of data so that needs to be fixed. Please include high quality figures that are clear and easy to see the details of the data
Response #1: Thanks for the reviewer’s kind suggestion. We have now provided figures of higher quality.
Point #2: the authors mention that " a drug library containing 1133 compounds .." What is this library? Where is it? Can it be included as a supplementary info? Where did this 1133 compounds come from?
Response #2: Thanks for the reviewer’s kind comments. We are sorry for the confusion. It is a commercial library (HY-LD-000001801, Med-Chem Express) that was used in the screening in our study. We have provided more information about this drug library in the Material and Methods. Also, the information of each compound has been listed in Supplementary Table S3.
Point #3: Any analytical HPLC after protein purification? Are the authors sure that the protein was purified to acceptable standards?
Response #3: Thanks for the reviewer’s kind comments. HPLC was used to analyze the level of c-di-AMP synthesized in the presence and absence of the inhibitor by ssDacA. Before HPLC analysis, the ssDacA protein was denatured by incubation at 100 °C for 10 minutes and then removed from the reaction (please see Section 4.5). So, only the supernatant was subjected to analysis by HPLC. We think that the SDS-PAGE data shown in Figure 1A suggests that the purifity of ssDacA (the last lane) is acceptable for the enzymatic reaction.
Point #4: Bacterial growth inhibition assay: The authors need to include (at least cite) the full anti bacterial assays protocol! Additionally even though the authors mention assays against B. subtilis, this bacterial strain is not included in the protocol/methodology section.
Response #4: Thanks for the reviewer’s kind comments. We have now included the detailed protocol for the bacterial growth inhibition assay (Please see Section 4.6). B. subtilis strain information has also been provided.
Point #5: It would be good if the authors could include a paragraph at the end of the conclusions section that suggests what would the future work include. There are so many papers on antimicrobial agents but a lot of them seem to be a dead end. It would be good if the authors could highlight a bit further what would the future work include? Why are these data more important than any other submitted papers?
Response #5: Thanks for the reviewer’s kind comments. We think that there are several pieces of work we can do in the future to make a lead compound to be more close to a drug. For example, structural biological studies can be performed to give a more accurate binding model between IPA-3 and ssDacA. It is also worth testing the bioactivity of IPA-3 derived compounds which can provide more information for structure-activity relationship analysis. If more promising compounds are discovered, druggability studies such as the absorption, distribution, metabolism, and excretion (ADME) analysis can be carried out. We have included this at the end of the Discussion section (Please see lines 266-273).

Round 2
Reviewer 3 Report
Despite the methodology and results do not present quantum mechanical calculation, the manuscript has merit and was successfully improved to attend the Journal audience. Classical methods reported sufficient results and can be a valuable at this point.